# Turtles in Malaysia: A Review of Conservation Status and a Call for Research

**DOI:** 10.3390/ani12172184

**Published:** 2022-08-25

**Authors:** Mohd Hairul Mohd Salleh, Yuzine Esa, Sarahaizad Mohd Salleh, Shahrul Anuar Mohd Sah

**Affiliations:** 1Department of Aquaculture, Faculty of Agriculture, Universiti Putra Malaysia, Serdang 43400, Selangor, Malaysia; 2Royal Malaysian Customs Department, Persiaran Perdana, Presint 2, Putrajaya 62596, Selangor, Malaysia; 3International Institute of Aquaculture and Aquatic Sciences, Universiti Putra Malaysia, Lot 960 Jalan Kemang 6, Port Dickson 71050, Negeri Sembilan, Malaysia; 4School of Biological Sciences, Universiti Sains Malaysia, George Town 11800, Penang, Malaysia

**Keywords:** taxonomic, sea turtles, IUCN Red List, CITES, checklist, genetic, endangered

## Abstract

**Simple Summary:**

Turtles are threatened all over the world. Malaysia has 24 species of turtles. This review focuses on current conservation status and some requirements for sustainability. We propose integrating concepts of ecology and molecular biology to provide almost comprehensive turtle reviews in Malaysia.

**Abstract:**

Approximately 356 species of turtles inhabit saltwater and freshwater habitats globally, except in Antarctica. Twenty-four species of turtles have been reported in Malaysia, four of which are sea turtles. The state of Terengganu harbored the highest number of turtles, with 17 different reported species. Based on the IUCN Red List, 29% of turtle species in Malaysia are critically endangered. In comparison, another 25% are classified as endangered. Likewise, CITES reported that 67% of Malaysia’s turtles are threatened, while 25% are classified as critically endangered. This review discusses the checklists, molecular genetics work, conservation status, recent trends, and recommendations for future research. Factors contributing to their population declines and current endangered status are also discussed.

## 1. Introduction

There are approximately 356 turtles living on land on every continent, except for in Antarctica, as well as in salt water and fresh water [1]. The term “turtle” is frequently used to refer to sea turtles, which only rarely leave the sea [2]. Sea turtles belong to the Cheloniidae families, except for the Leatherback turtle, which is the only genus in the Family Dermochelyidae and has a leathery carapace [3]. The seven species of sea turtles are Green turtle (*Chelonia mydas*), Hawksbill turtle (*Eretmochelys imbricata*), Leatherback turtle (*Dermochelys coriacea*), Loggerhead turtle (*Caretta caretta*), Flatback turtle (*Natator depressa*), Olive ridley turtle (*Lepidochelys olivacea*), and Kemp’s ridley turtle (*Lepidochelys kempii*) [4]. Malaysia is home to four sea turtle species: the Leatherback turtle, the Green turtle, the Olive ridley turtle, and the Hawksbill turtle [5]. With nearly 40% of its total body mass made up of bone, the turtle is possibly the most organized form of animal armor ever to appear [6]. As a result, this great armor is most likely why turtles appeared on the scene over 200 million years ago and miraculously survived the extinction of the dinosaurs and other devastating events [6].

In Testudines’ order, turtles are any reptile with a hard shell around its body, including tortoises [7]. They have anatomical characteristics that set them apart from other turtles. In the Chelonia order, a turtle, a tortoise, and a terrapin are all names for hard-shelled egg-laying reptiles [8]. However, the specific expression used for a particular turtle can vary depending on its natural surroundings. For example, the term “turtle” usually refers to turtles that have spent their entire lives in or near water [9]. The term “tortoise” is commonly used to refer to turtles that spend most of their time ashore, eating bushes, grass, and fruit [10]. Unlike other turtle family members, tortoises do not have webbed feet because they do not spend much time in the water [11]. Terrapins are turtles that invest energy in fresh and brackish water [12]. “Terrapin” is derived from an Algonquian Indian word that means “a small turtle” [13]. Malaysia is home to 20 different kinds of turtles. Two of them, *Pelodiscus sinensis* and *Trachemys scripta*, were brought there from other places [14]. In total, Malaysia has 24 different species of turtles [15].

The most established bodies in conservation biodiversity are the IUCN (International Union for Conservation of Nature) and CITES (Convention on International Trade in Endangered Species of Wild Fauna and Flora). The IUCN Red List is a rich compendium of information on threats, ecological requirements, habitats of species, and conservation actions that can be taken to prevent extinctions [16,17]. It is a target framework for evaluating the extinction risk of species depending on past, present, and extended threats [18]. The assessments are carried out by following a standardized procedure that employs the rigorous IUCN Red List Categories and Criteria, ensuring the highest scientific documentation standards, information management, expert review, and justification [16,19]. More assessments will aid in developing the IUCN Red List as a completely comprehensive “Barometer of Life” [20]. Threatened species include those listed as critically endangered, endangered, or vulnerable [21].

In contrast, CITES is an international agreement among governments [22]; it ensures that international trade in wild animals and plant specimens does not threaten their numbers [23]. Due to the international nature of the trade in wildlife, its regulation requires international cooperation to protect certain species from over-exploitation [24].

Today, it protects more than 37,000 plant and animal species in different ways, depending on whether they are traded either as dried herbs, live specimens, or fur coats [25] CITES protects approximately 5950 animals and 32,800 plant species from over-exploitation through international trade [26]. They are listed in the three CITES Appendices [27]. The species are organized in the appendices based on their vulnerability to the global trade.

Therefore, the turtle checklist in Malaysia was presented previously, but the current conservation status needs to be updated. Thus, we provide the species checklist of turtles in Malaysia, document the current status of turtles in Malaysia, and compile the turtle research trends nowadays. We also review the causes of current precipitous declines in turtle populations by examining the threatened status of turtles in Malaysia and speculate on the potential reasons for extinction in the coming century. In doing so, we make a list of common threats and describe the research that needs to be performed in order to ensure long-term survival.

## 2. Turtle Types in Malaysia

The “hard-shelled” Cheloniidae advanced around 60 million years ago, and the “delicate-shelled” Dermochelyidae developed approximately 90 million years ago [28,29]. The Cheloniidae contain six surviving species in five genera: the Flatback turtle (*Natator depressus*), the Green turtle (*Chelonia mydas*), the Hawksbill turtle (*Eretmochelys imbricata*), the Loggerhead turtle (*Caretta caretta*), the Kemp’s ridley turtle (*Lepidochelys kempii*), and the Olive ridley turtle (*Lepidochelys olivacea*). The Leatherback turtle (*Dermochelys coriacea*) is the only extant species in the Dermochelyidae family [30].

There are seven sea turtle species (Cheloniidae and Dermochelyidae) worldwide, with five nesting in Southeast Asia. Some of these species are shown in Appendix A, and Figure 1 shows that 17% of the studied sea turtles can be found locally. The other two species are the Kemp’s ridley turtle, which only lives in the Western Atlantic Ocean, and the Flatback turtle, which only lives in Australia (though its feeding grounds extend into Eastern Indonesia) [31].

Testudinidae (Land Tortoises): 60 land tortoise species are recognized worldwide, accounting for 8% of the 24 turtle species in Malaysia (Figure 1). Most of them live in semi-arid, dry open habitats, including grasslands and deserts [32]. Only five Southeast Asian species have adapted to different environments, including humid, forested habitats, and cooler temperatures in a lower montane forest [31,33]. Appendix A shows some examples.

Asian hard-shell turtles (Geoemydidae) have the most turtle family species, with approximately 40 species of 14 genera found in Southeast Asia and 70 species found worldwide. Malaysia has the most of this turtle type (46%) (Figure 1). DNA sequencing has recently revealed hidden diversity in this group; for example, the Cyclemys leaf turtles are now classified as six distinct species [31]. Appendix A shows some examples.

The big-headed turtle (*Platysternon megacephalum*) is mainly the only individual from this family, Platysternidae; it is remembered for the superfamily Testudinoidea, which likewise incorporates the Testudinidae (land turtles), Geoemydidae (Asian hard-shelled turtles), and Emydidae (new world reptiles). This turtle has a large head that it cannot retract into its shell [34].

Softshell turtles (Trionychidae) have 30 species worldwide [35], with 15 species in Southeast Asia and six found in Malaysia (Appendix A). They also have a flexible, rugged carapace [36], and Southeast Asia is home to approximately half of the world’s softshell turtles [31,33].

The snake-necked turtles of the Chelodina family are an ancient group of expert fish-eaters whose long necks must be turned sideways to reach beneath the carapace [37]. There are 16 species in the world under this family [31,38].

The pig-nosed turtle (*Carettochelys insculpta*) is the only species in the Carettochelyidae family [39]. It is found in just three nations—Indonesia, Papua, New Guinea, and Australia [40]. These species are kept as pets in Malaysia. Their flippers resemble those of sea turtles, and their carapace is rough, but their most unique feature is their pig-like nose [31,41].

Turtles from the Emydidae family originated in the Americas and are widely available in the world as the most popular pet [42]. For instance, consider the Red-Eared Slider [31].

The turtles in the family Emydidae belong to the order Testudines and the suborder Cryptodira. There are about 52 species in this family, which is divided into 12 genera: *Actinemys*, *Chrysemys*, *Clemmys*, *Deirochelys*, *Emydoidea*, *Emys*, *Glyptemys*, *Graptemys*, *Malaclemys*, *Pseudemys*, *Terrapene*, and *Trachemys* [43]. Except for *Trachemys*, which is found in South America and the West Indies, and *Emys*, which inhabits Southern Europe, Northern Africa, and Western Asia, all of these species are restricted to North America [44]. The relationships between the 12 genera and the species that make up the family are yet unknown [45].

## 3. Molecular Research Trends

### 3.1. Mating System

In turtles, molecular markers have been used to describe the mating system in the existing species, with multiple paternity and sperm storage being distinguishing features [46,47,48,49]. In contrast, the IUCN lists the Hawksbill turtle (*Eretmochelys imbricata*) as critically endangered [50,51]. *Eretmochelys imbricata* was shown to be primarily monogamous in two studies [52,53] so that sperm could be stored [53]. Nonetheless, one of these examinations was conducted in an isolated area of Malaysia [52]. Moreover, the other one was found in a remote part of the Republic of Seychelles [53]. It has been suggested that different sea turtles may have different levels of paternity depending on where they live and how abundant there are [47,48,54].

### 3.2. Population Genetics Analysis

Table 1 briefly describes genetic markers and how they were used in turtle studies. Prior studies have also used multilocus minisatellites (“DNA fingerprinting”) [55]. Microsatellites, along with arbitrary intensified polymorphic DNA (RAPDs) [56], allozymes [57,58,59,60], and the anonymous single-copy nuclear DNA (scnDNA) [61,62,63,64,65], are the most commonly used markers for paternity analysis. Other studies have used microsatellites to analyze population genetics [46,64,66,67,68,69,70,71,72] and molecular evolution [73,74]. A few were reported to be the development of new markers [75,76] or developed attributes for species identification [77]. Microsatellite markers were compared to mtDNA haplotype markers, which are passed down from the mother and have been used for a long time to study the genetic structure and phylogeography of turtle populations [48,67,68].

### 3.3. Genetic Variability

Due to molecular markers in various fields of knowledge, there has been a significant advancement in molecular genetics techniques in recent years. These markers are practical and risk-free tools that can facilitate an accurate diagnosis. Comparative studies of the genetic variation described by molecular markers and regional morphometric patterns have centered on several wild turtle species [78]. The following molecular markers (Table 1) stand out among those used to detect genetic variability in DNA sequences, such as restriction fragment length polymorphism (RFLP), random amplified polymorphic DNA (RAPD), single nucleotide polymorphism (SNP), mitochondrial DNA (mtDNA), and microsatellites or short tandem repeats (STR) [79]. They stand out because they provide ecological and morphological data that can be used to make different plans for protecting most genetic resources [48].

For set preservation systems, molecular markers help us to better understand the genetic variety of faunal species and segment relations. The low genetic variety and star-shaped haplotype network of sea turtles in the Eastern Pacific suggest that the species most likely originated in the Indo-Pacific on a generally late transformative timescale [80,81]. Most sea turtles can be found all over the world, meaning that natural damage and environmental loss will probably affect their movements [82,83], assuming that these effects reduce availability and population size. If that happens, genetic diversity will be lost, and this will increase the risk of extinction [84] caused by potential inbreeding depression, which makes animals less able to change and reduces fitness [85].

The rate at which genetic variability is lost in specific animal varieties is a component of the successful population size. This boundary is impacted by an organic entity’s characteristic history and segment history [85]. An assessment of crucial boundaries that shape compelling population size is expected to measure the danger of hereditary disintegration and inbreeding depression [86]. The principal is information on both quality streams via an animal type’s geographic appropriation and the mating framework. The mating framework fact is that, when the successful population’s size is less than the registered size, the regenerative slant is high [87].

**Table 1 animals-12-02184-t001:** Genetic markers for molecular turtle research.

Allozymes	Alleles of a catalyst can be identified by various electrophoretic versatility (usually starch or cellulose acetic acid derivation gels). Since mtDNA haplotypes and microsatellites have become more popular as markers for population genetics and paternity analysis [48,57,88], a well-developed method has been used only a few times in turtle concentrates.
Anonymous scnDNA	Various PCR preliminaries have been created for sea turtles to intensify mysterious single-copy nuclear DNA loci. Variety in intensified items is inspected by RFLP or sequencing and has not been utilized much since microsatellites became the more mainstream nuclear DNA marker [48,61,62,63,64,65,89].
Microsatellites	Pair rehashes of a 1–6 bp “core” grouping. The evaluation strategy gives single-locus data. Bespoke introductions for PCR intensification are intended for the microsatellite’s flanking areas. The hypervariable idea of a microsatellite is several rehashes of the left change effectively. Moreover, single-locus information implies that more impressive scientific strategies are conceivable than multilocus fingerprinting. It has become a mainstream marker for the population’s genetic qualities. It is a technique for determining paternity in turtles [48,90,91,92].
Minisatellites	This is the first DNA fingerprinting technique. Dispersed across the nuclear genome are families of tandemly repetitive “minisatellite” regions sharing a 10–15 bp “core” sequence. Variation in the recurrent number is acquired. Moreover, minisatellites are exceptionally polymorphic (hypervariable) as genetic markers at the individual and population levels. It has been used only once for sea turtles since microsatellites became the more common marker [55,93].
MtDNA Haplotypes	For sea turtles, it is normal to utilize arrangement variety in the control district of mitochondrial DNA (mtDNA). These are named “haplotypes” because the mitochondrial genome exits as a single copy (haploid). An approximately 400 bp piece is enhanced with a standard arrangement of groundworks. In current examinations, the variety is composed of the improved items by sequencing. More seasoned investigations may utilize Restriction Fragment Length Polymorphism (RFLP) or other fast screening methods. The haplotypes at the Archie Carr Center for Sea Turtle Research are put in order by a normalized classification [94,95].
RAPDs	Random Amplified Polymorphic DNA (RAPD). A PCR-based procedure utilizes short (10 bp) oligonucleotide primers in random arrangement to create different PCR results of contrasting sizes isolated on an agarose gel. The multilocus technique was momentarily famous in molecular ecology because of its modest and straightforward convention. However, it tumbled from favor when reproducibility turned into an issue. I am mindful of just one turtle study that utilized it [56,96,97].
RFLP	Restriction Fragment Length Polymorphism (RFLP). Limitation compounds have the potential to cut DNA at explicit acknowledgement groupings. In order to create parts of reproducible size from any substrate DNA particle (nuclear or mitochondrial DNA), they should be be isolated by the electrophoresis process. The variation between individuals in the sizes of DNA fragments is caused by mutations that create or eliminate restriction enzyme recognition sequences and is used as an indication of genetic variation. It was previously popular for assessing mtDNA haplotype variation in turtles, but direct sequencing has largely replaced RFLP. In addition, it was briefly used in turtle studies for anonymous scnDNA [48,61,62,63,64,65,98,99].

## 4. Conservation Status

### 4.1. The Status of the IUCN Red List

The IUCN Red List categorizes species into nine groups (Table 2), which Reference [100] defined based on population size, rate of decline, geographic distribution area, fragmentation distribution, and population degree. The importance of applying any measures without extensive information, including suspicion and potential future threats, is emphasized “so long as these can reasonably be supported” [19]. The “Threatened” category includes “Critically Endangered”, “Endangered”, and “Vulnerable” [21] on its Red List.

Table 3 shows that Malaysia has 24 turtle species, four of which are sea turtles, and the other 20 are freshwater turtles (two of which are introduced species) [14,101]. According to the IUCN Red List, a sea turtle (*Eretmochelys imbricata*) and six freshwater turtle populations (*Manouria emys*, *Batagur affinis*, *Orlitia borneensis*, *Batagur borneoensis*, *Indotestudo elongata*, and *Chitra chitra*) are critically endangered in Malaysia (Figure 2). In contrast, a sea turtle (*Chelonia mydas*) and five freshwater turtles (*Heosemys annandalii*, *Cuora amboinensis*, *Heosemys spinosa*, *Chitra indica*, and *Pelochelys cantorii*) were endangered in Malaysia. Two sea turtles (*Dermochelys coriacea* and *Lepidochelys olivacea*) and six freshwater turtles were vulnerable (*Malayemys macrocephala*, *Notochelys platynotan*, *Siebenrockiella crassicollis*, *Amyda cartilaginea*, *Manouria iimpressa*, and *Pelodiscus sinensis*). However, two sea turtles were reported by Reference [102] in *The ASEAN Post*; a source from the World Wildlife Foundation (WWF) Malaysia shows that the Leatherback turtle is critically endangered, and the Olive Ridley turtle is endangered in the Malaysian ocean. Moreover, one species, *Cyclemys dentata*, is near threatened, and two species, *Dogania subplana* and *Trachemys scoundaripta*, are less concerned.

All the turtle species are distributed all over Malaysia. However, Terengganu is home to 17 species, including four species of sea turtles (*Chelonia mydas*, *Dermochelys coriacea*, *Lepidochelys olivacea*, and *Eretmochelys imbricata*) and 13 species of freshwater turtles (*Trachemys scripta*, *Batagur affinis*, *Batagur borneonsis*, *Coura amboinensis*, *Siebenrockiella crassicollis*, *Manouria emys*, *Amyda cartilaginea*, *Dogania subplana*, and *Pelochelys cantorii*) [14]. In addition, referring to Figure 3, the IUCN Red List analysis shows that 29 percent of Malaysia’s turtle species are critically endangered and 25 percent are endangered.

**Table 2 animals-12-02184-t002:** The IUCN Red List classifies species into nine groups [19,100,103].

Classification	Describtion
Not evaluated (NE)	Not yet assessed by the IUCN, they indicate species that have not been reviewed enough to be assigned to a category.
Data deficiency (DD)	Offering insufficient information for a proper assessment of conservation status to be made.
Least concern (LC)	It is unlikely to become extinct soon.
Near threatened (NT)	Close to being at an increased risk of extinction soon.
Vulnerable (VU)	It is considered at an increased risk of unnatural (human-caused) extinction without further human intervention.
Endangered (EN)	A very high risk of extinction in the wild.
Critically endangered (CR)	Points in a particular and extremely critical state.
Extinct in the wild (EW)	Point only lives on in zoos, farms, and places outside of its native range, as surveys have shown.
Extinct (EX)	Beyond a reasonable doubt, the species is no longer extant.

**Table 3 animals-12-02184-t003:** Checklist of Turtle Species in Malaysia [21,104].

Common Name	Scientific Name	GenBank Accession	IUCN Red List Status	CITES Appendix	Reference
Asian Narrow Headed Softshell Turtle	*Chitra chitra*	HQ329770	CR	I	[105]
Hawksbill Turtle	*Eretmochelys imbricata*	GQ152887	CR	I	[71]
Southern River Terrapin	*Batagur affinis*	MN069310	CR	I	[106]
Asian Giant Tortoise	*Manouria emys*	KP268838	CR	II	[107]
Elongated Tortoise	*Indotestudo elongata*	KP268857	CR	II	[108]
Malaysian Giant Turtle	*Orlitia borneensis*	HQ329693	CR	II	[105]
Painted Terrapin	*Batagur borneoensis*	HQ329672	CR	II	[105]
Green Turtle	*Chelonia mydas*	MN124278	EN	I	[109]
Asian Giant Softshell Turtle	*Pelochelys cantorii*	HQ329785	EN	II	[105]
Indian Narrow-headed Softshell Turtle	*Chitra indica*	HQ329771	EN	II	[105]
Malaysian Box Turtle	*Cuora amboinensis*	JN860217	EN	II	[108]
Spiny Turtle	*Heosemys spinosa*	HQ329684	EN	II	[105]
Yellow-headed Temple Turtle	*Heosemys annandalii*	HQ329681	EN	II	[105]
Leatherback Turtle	*Dermochelys coriacea*	KU883273	VU	I	[110]
Olive Ridley Turtle	*Lepidochelys olivacea*	KF894766	VU	I	[111]
Asiatic Softshell Turtle	*Amyda cartilaginea*	HQ329768	VU	II	[105]
Black Marsh Turtle	*Siebenrockiella crassicollis*	HQ329704	VU	II	[105]
Impressed Tortoise	*Manouria impressa*	GQ867670	VU	II	[112]
Malayan Flat-shelled Turtle	*Notochelys platynota*	HQ329692	VU	II	[105]
Malayan Snail-eating Turtle	*Malayemys macrocephala*	HQ329686	VU	II	[105]
Chinese Softshell Turtle	*Pelodiscus sinensis*	JQ844545	VU	None	[113]
Asian Leaf Turtle	*Cyclemys dentata*	HQ329676	NT	II	[105]
Malayan Softshell Turtle	*Dogania subplana*	NC_002780	LC	II	[114]
Yellow-bellied Slider Turtle	*Trachemys scripta*	JF700194	LC	None	[115]

**Figure 2 animals-12-02184-f002:**
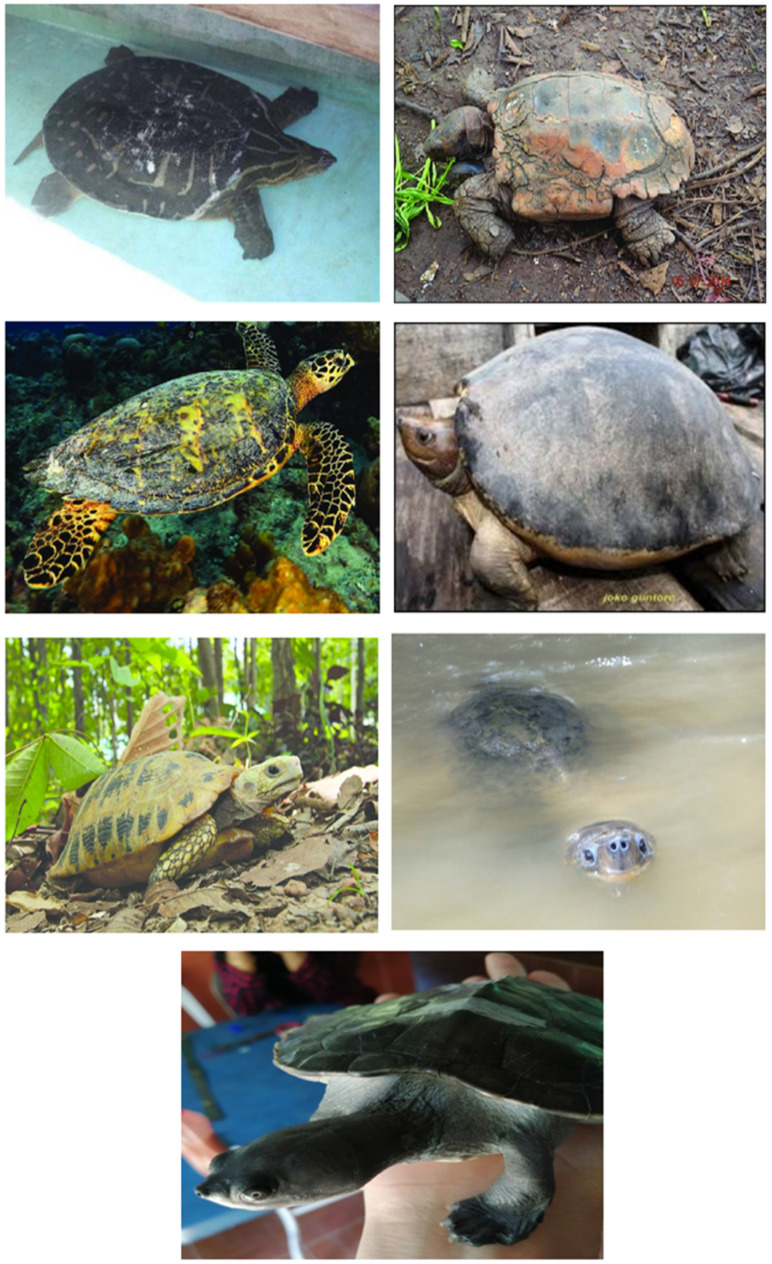
The critically endangered turtles in Malaysia. Top left to right: *Chitra chitra* [116], *Manouria emys* [117], *Eretmochelys imbricata* [118], *Batagur borneensis* [119], *Indotestudo elongata* [120], *Orlitia borneensis* [121], and *Batagur affinis* [122].

**Figure 3 animals-12-02184-f003:**
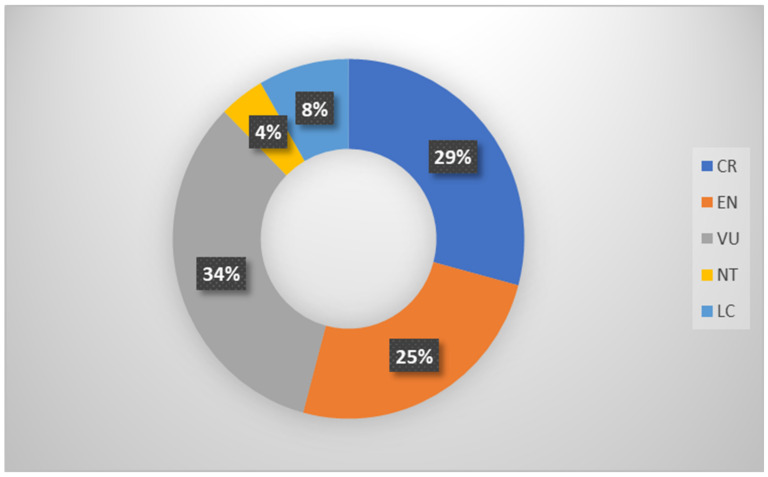
Chart of IUCN Red List status on turtles.

### 4.2. The CITES Appendices

The Convention’s Appendices I, II, and III are lists of species with different levels of protection from over-exploitation [123]. Appendix I lists the most endangered plants and animals on the CITES list. They are almost extinct, but CITES allows international trade in specimens of these species as long as the import is not for commercial use (i.e., a scientific research study) [124].

In Appendix II, there is a list of species that are not threatened with extinction right now, but if the trade is not controlled, there is a high chance that they will be in the future. It also includes supposed “similar species”, such as species whose standards in exchange resemble species recorded for conservation purposes. Trade-in specimens of Appendix-II species may be authorized by issuing an export permit or re-export permit certificate. No import permit is necessary for these species under CITES (although a permit is needed in some countries with stricter measures than CITES requires) [125].

Appendix III contains a list of species added at the request of a party that already regulates international trade in the species. Specimens of the species in this appendix can be traded around the world only if the proper permits or certificates are shown [123].

An analysis of Figure 4 reveals that CITES has classified Malaysian turtles as 67 percent threatened, including *Manouria emys*, *Orlitia borneensis*, *Batagur borneoensis*, *Indotestudo elongate*, *Heosemys annandalii*, *Cuora amboinensis*, *Heosemys spinosa*, *Chitra indica*, *Pelochelys cantorii*, *Malayemys macrocephala*, *Notochelys platynotan*, *Siebenrockiella crassicollis*, *Amyda cartilaginea*, *Manouria impressa*, *Cyclemys dentata*, and *Dogania subplana*. About 25% (*Batagur affinis*, *Chitra chitra*, *Eretmochelys imbricata*, *Chelonia mydas*, *Dermochelys coriacea*, and *Lepidochelys olivacea*) are the most endangered species.

## 5. Threat Factors

The [21] population trends for all VU, EN, and CR turtle populations are decreasing. In conclusion, many factors contribute to threats. This review article compiles and documents the work of other Malaysian researchers and decision-makers for future reference (Figure 5). The primary causes of concern are egg consumption and trade [4,126]. The main threats to turtles are illegal and unregulated turtle poaching by Hainan (China) vessels and Vietnam [1]. Turtles are hunted for food, medicine, and ornaments [127]. In Malaysia, religious beliefs have reduced the killing of adult turtles for food [128]. On the other hand, building dams, taking turtle eggs, removing riparian vegetation, sand mining, and drowning in fishing nets are some of the turtle’s most significant problems [129,130,131,132,133].

According to References [134,135], the most critically endangered turtle species may become the most sought after due to their scarcity, which makes them especially valuable in the pet trade, hunting, and habitat degradation. Reference [136] reports that they are eaten, collected, butchered, and traded in large numbers; they are used for pets, food, and traditional medicine—eggs, juveniles, adults, and body parts are all exploited indiscriminately, with no regard for sustainability [137]. Their habitats are being destroyed, developed, fragmented, and polluted at an alarming rate [138,139]. Species all over the world are threatened or vulnerable, with many critically endangered. Others are on the verge of extinction, and a few have already perished [140]. Humans are threatening the extinction of countless eons and turtles [141].

Aside from overt and highly impactful conservation threats such as overexploitation and habitat destruction, the global turtle fauna is also increasingly facing another insidious threat: genetic pollution caused by human-facilitated hybridization and introgression from introduced and invasive species [142,143,144,145,146]. Although it is not entirely new, the current scale is unprecedented. Some taxa have already been impacted in the past. This is most likely true for Pelodiscus Asian softshell turtles. These turtles have been farmed and traded for hundreds of years. As a result, different species and local genetic lineages have been moved, leading to other taxa and lineages in captivity and the wild [147,148].

Similarly, the historical introduction of *Mauremys reevesii* to Japan resulted in massive hybridization with the native [149]. Another historical case of human-mediated admixture of genetic lineages is known from European pond turtles (*Emys orbicularis*). The non-native populations on the Balearic Islands, which were most likely introduced during Roman times [150], are of admixed origin [151]. Another population with genetic signatures of an old or ancient introduction of *Emys orbicularis hellenica* was discovered near Rome [151,152] within the range of another subspecies (*Emys orbicularis galloitalica*). However, unlike in the past, when only a few turtle species were affected, genetic pollution has become a big problem in protecting wildlife in recent years. This is because of the huge pet and food trade and increased human mobility.

Today, genetic pollution is also caused by well-meaning augmentation of endangered local turtle populations with genetically mismatched individuals (typically, but not exclusively, from non-coordinated actions by turtle enthusiasts), the release of surplus or abandoned genetically divergent pet turtles, and also by large-scale releases of confiscated turtle shipments, especially in Southeast Asia. Some endangered *Emys orbicularis* populations are on the northern edge of their range [153,154], and there is genetic evidence for restocking with multiple subspecies; in southern France [152,155], there is evidence of restocking with non-native *Emys orbicularis hellenica* rather than native *Emys orbicularis.* Examples of genetic pollution caused by abandoned pet turtles include *Chrysemys picta bellii* from British Columbia, introgressed by non-native subspecies [49], and Antillean (*Trachemys*), introgressed by Red-Eared Sliders (*Trachemys scripta elegans*) [156]. As previously stated, some cases involving European pond turtles are related to genetic contamination caused by abandoned pet turtles. In Taiwan, hybridization between *Mauremys reevesii* and *Mauremys sinensis* has been observed in the wild in released trade animals [157]. According to Reference [158], preserving well-defined genetic lineages, subspecies, and species that are mostly pure and not hybridized is critical. Therefore, in Malaysia, the two introduced species potentially cause genetic pollution.

## 6. Future Research and Recommendations

Several commendable research lines would also aid in understanding the growing issues and threats driving turtles to extinction. Table 4 discusses and summarizes these points.

To distinguish explicit from common-sense preservation activities, it is necessary to examine each species’ information and protection status [173]. Protection status is frequently based on IUCN-proposed measures, which consider factors such as the number of the population remaining, the size of the current geographic conveyance, and data on idle dangers [19]. The restricted assets are given for turtle preservation in many countries, including Malaysia. Reference [174] says that it is occasionally necessary to have a target method for focusing on species. We believe that our work will help to collect more reliable data on turtles and address the threats to Malaysia’s most vulnerable species. Our research can help determine which species require management plans and investment from government agencies. Moreover, the research priorities can help researchers and students figure out which topics have not been studied sufficiently but are more important for making conservation recommendations [175]. The author of Reference [176] says that it is essential to direct human and economic efforts toward those issues that are high priorities. They also say that governments and regulatory authorities should work together to ensure that laws and rules are enforced more effectively [177,178].

It is critical to encourage short-to-medium-term studies in ecology and molecular systematics, as well as medium-to-long-term demographic studies that are biologically representative and can guide conservation and management efforts. Malaysian continental turtle species face significant research challenges. They entail leaving our geographical and thematic comfort zones to face the difficulty of gaining access to lesser-known species. Developing studies necessitate significant effort over long periods of time. We should bring together a wide range of expertise, such as taxonomists, ecologists, and molecular biologists. Those research outputs may directly raise public awareness of turtles’ plight toward sustainability.

## 7. Conclusions

In a small number of species, we face a turtle endurance crisis that is both severe and urgent. Without intervention, valuable species will become extinct over the next few decades. Some of the future research requirements listed in this review may assist researchers in collaborating to save these turtles for which we are so passionate. This review’s benefits may help readers understand the risks and threats to turtles and set goals for how long we should be able to last and how safe we should be, as well as making new standards for practical conservation techniques.

## Figures and Tables

**Figure 1 animals-12-02184-f001:**
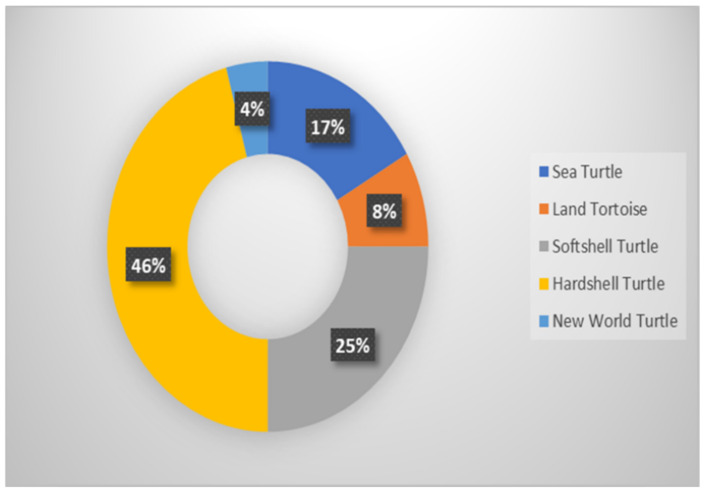
The diversity of turtles in Malaysia.

**Figure 4 animals-12-02184-f004:**
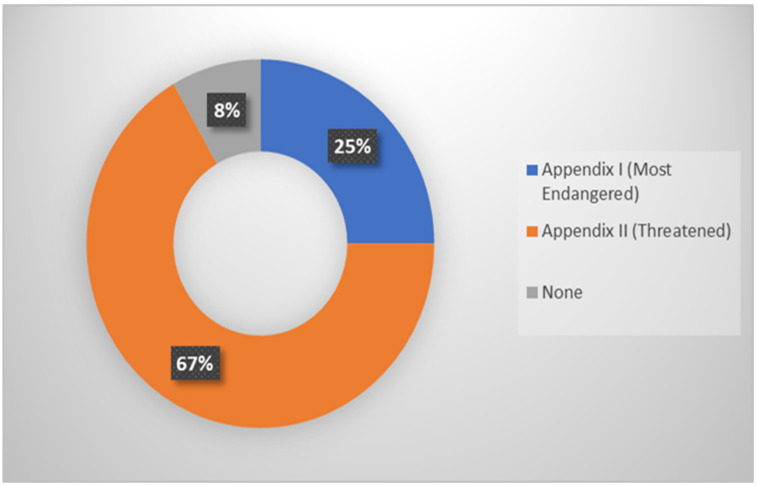
Chart of CITES’s appendices on turtles.

**Figure 5 animals-12-02184-f005:**
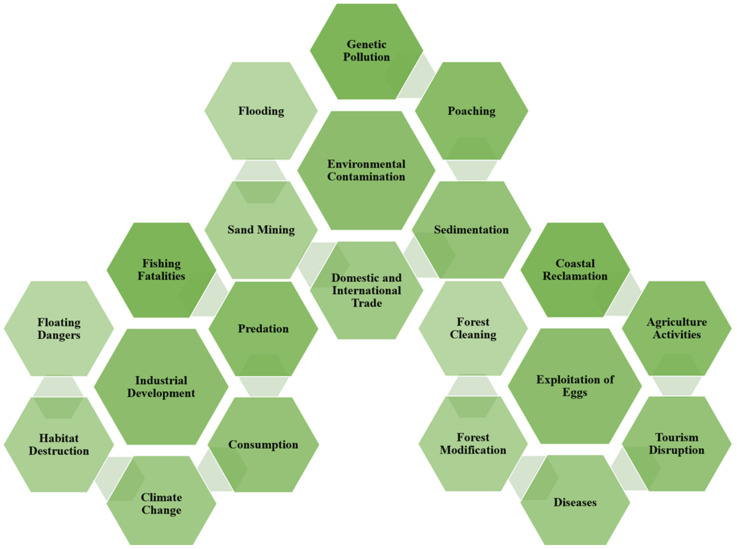
These threat factors were compiled from IUCN data, the DOF Report, DWNP Report, TRAFFIC South-east Asia Report, species-recovery plans, federal-agency re-sponses, and miscellaneous publica-tions on species’ life history. A complete list of documents used to assign biological attributes to endangered species is available from the authors.

**Table 4 animals-12-02184-t004:** A summary of recommended future research priorities.

Topic	Method
Ecosystem effects	Monitor key turtle habitats to generate baseline data. Mesocosm experiments team up with other research disciplines and industries. Create strategies to identify and measure the trophic exchange of plastic, related poisons, and bioaccumulation. Explore the effect of plastics on the cycle of benthopelagic coupling [159].
Impacts on nesting beaches	Record perceptions of experiences with seashore garbage for females and hatchlings. Use oceanographic demonstrating to conjecture how and when key waterfront regions are prone to being affected by plastic contamination [159].
New sites	The purpose of the examination is to recognise new nesting zones, especially if current nesting locales become unsatisfactory because of improvement or environmental-driven change [160,161].
Embryology	Developing and assessing a reliable indicator of hatchling health, comprehending endocrine influences on embryology, and further research into the role of home site selection in hatchling development [162].
Molecular	There are numerous ways to deal with understanding the spatial biology of turtles,
	counting hereditary qualities [163], natural biomarkers like stable isotopes [164].
Conservation Management	Designing management strategies with SMART (specific, measurable, achievable, realistic and time-based) objectives that permit assessment, variation, and the advancement of proof-based preservation will be critical to deciding the board achievement of current and future ventures [162].
Climate Change	Understanding cumulative impacts or developing conservation responses to climate change [165].
Threats	Thought of future dangers and their management in decision processes like horizontal planning [166]. GIS should provide new insights into patterns and can greatly aid in understanding the effects of hazards and the sufficiency of relief [167,168].
Habitat Restoration	The carrying capacity of territory is a significant consideration in living space reclamation [169]. A few researchers have attempted to explain the general or current-carrying capacities of specific biological systems [170,171], though much more work is needed here.
eDNA	Streamlining field techniques for turtle eDNA assortment, further testing primer explicitness through trials of tests containing numerous species’ DNA, and creating primers focusing on other turtle networks could extraordinarily improve the recognition rates of uncommon species [172].
nDNA	Nnuclear DNA markers (e.g., microsatellites, SNPs) are expected to confirm and further assess the hereditary portrayal of turtles in the EP as the information from mtDNA markers just reflects variety among female genealogies [47,70].
Microbiology	Harmful microorganisms such as viruses, bacteria, parasites, and fungus that have not yet been investigated on turtles through metabarcoding, which has the potential to spread among or between hosts. Aside from that, future research could look into the impact of the dominant phylum (Proteobacteria) and genus (*Cetobacterium*) [122].

## Data Availability

Not applicable.

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
