# Peer review of "Turtles in Malaysia: A Review of Conservation Status and a Call for Research"

_animals, 2022, doi:10.3390/ani12172184_

Round 1

Reviewer 1 Report

In this manuscript, the authors discussed the checklists, molecular genetics work, conservation status, recent trends, and recommendations for future research on turtles in Malaysia. There are some concerns need to be addressed before it is accepted.

1.      The order of the statements in Abstract section needs to be adjusted.

2.      Line 39: here, should be sea turtles.

3.      The third section “Molecular Research Trends” should be divided into subheadings such as molecular markers, phylogenetic evolution et al.

4.      Please check the case of figure legends and table notes of the full text.

Reviewer 2 Report

The concept for this article is very good and the publication will be very useful when completed.  The work reviews and combines several important studies.  However, in many cases I was not sure of the meaning. 

This reviewer did find the writing somewhat difficult to understand.  I recommend the authors develop a collaboration with someone who has more experience in writing scientific English.  In many cases I was not sure of the meaning.  I have tried to provide a few edits in the attached copy that I hope will be useful to the aurthors.

Round 2

Reviewer 2 Report

There is still some confusion on the use of the terms terrapin versus turtle.  My understanding is that only the two Batagur species are properly called terrapins as well as the North American Diamondback terrapin.   Please double check this.  I was also a little confused in your discussion of the softshell turtles.  You seem to say they also have hard shells. 

Otherwise the manuscript is improved and please see a few suggested wording changes in the attached copy. 

Author Response

The amendment has been made following your suggestion. Thank you for bringing up some issues. A more clear statement and sentences have been made.